# Leaf Extracts from Resistant Wild Tomato Can Be Used to Control Late Blight (*Phytophthora infestans*) in the Cultivated Tomato

**DOI:** 10.3390/plants11141824

**Published:** 2022-07-12

**Authors:** Ramadan A. Arafa, Said M. Kamel, Dalia I. Taher, Svein Ø. Solberg, Mohamed T. Rakha

**Affiliations:** 1Plant Pathology Research Institute, Agricultural Research Center, Giza 12619, Egypt; arafa.r.a.85@gmail.com (R.A.A.); said_kamel88@yahoo.com (S.M.K.); 2Vegetable Crops Research Department, Horticulture Research Institute, Agriculture Research Center, Giza 12619, Egypt; daliataher1981@gmail.com; 3Faculty of Applied Ecology and Agricultural Sciences, Inland Norway University of Applied Sciences, 2418 Elverum, Norway; 4Department of Horticulture, Faculty of Agriculture, University of Kafrelsheikh, Kafr El-Sheikh 33516, Egypt

**Keywords:** late blight, *Phytophtora infestans*, resistance mechanism, tomato, volatile organic compounds

## Abstract

Late blight disease, caused by *Phytophthora infestans* (Mont.) de Bary, is one of the most challenging diseases threatening tomato production and other Solanaceae crops. Resistance to late blight is found in certain wild species, but the mechanism behind the resistance is not fully understood. The aim of this study was to examine the metabolic profiles in the leaf tissue of late blight-resistant wild tomato and to investigate if leaf extracts from such genotypes could be used to control late blight in tomato production. We included three recognized late blight-resistant wild tomato accessions of *Solanum habrochaites* (LA1777, LA2855, and LA1352) and two recognized highly susceptible genotypes, *S. lycopersicum* (‘Super Strain B’) and *S. pimpinellifolium* (LA0375). The metabolic profiles were obtained in both inoculated and non-inoculated plants by analyzing leaf extracts using high-resolution gas chromatography-mass spectrometry (GC-MS) with three replicate analyses of each genotype. We focused on volatile organic compounds (VOCs) and identified 31 such compounds from the five genotypes with a retention time ranging from 6.6 to 22.8 min. The resistant genotype LA 1777 produced the highest number of VOCs (22 and 21 in the inoculated and control plants, respectively), whereas the susceptible genotype ‘Super Strain B’ produced the lowest number of VOCs (11 and 13 in the respective plants). Among the VOCs, 14 were detected only in the resistant genotypes, while two were detected only in the susceptible ones. In vitro trials, with the use of a detached leaflet assay and whole-plant approach, were conducted. We revealed promising insights regarding late blight management and showed that metabolic profiling may contribute to a better understanding of the mechanisms behind *P. infestans* resistance in tomato and its wild relatives.

## 1. Introduction

Tomato (*Solanum lycopersicum* L.) is the most commonly grown vegetable crop after potato (*S. tuberosum* L.) [1]. According to the FAOSTAT [2], the global cultivated area of the tomato is approximately five million hectares with an annual production of around 187 million tonnes. China, USA, India, Turkey, and Egypt are the top five tomato-producing countries. Tomato is cultivated in greenhouses, house gardens, and open fields [3] and is a good provider of vitamins (A, B, C, and K), minerals (Cu, K, Mn, Mg, P, Fe, and Zn) and other micronutrients that can act as an antioxidant to help prevent diseases [4]. Tomato can grow in a broad spectrum of environmental conditions, it has a relatively short life cycle, hybridizations are simple, and cultivars with diverse characteristics are available. The plant is susceptible to various diseases, including bacteria, fungi, viruses, and viroids [5]. One of the most serious diseases is the late blight, caused by *Phytophthora infestans* (Mont.) de Bary. Late blight has led to significant economic losses in the production of tomato and potato crops, and has caused global food security concerns [6]. *P. infestans* is an oomycete pathogen that is highly destructive and has the capacity to destroy tomato plants within 7 to 10 days under cool and humid conditions [7,8]. Farmers apply intensive fungicide spraying to avoid the disease. An alternative and a more environmentally friendly method is resistance breeding.

To date, six late blight resistance genes/genomic regions have been identified (*Ph-1, Ph-2, Ph-3, Ph-4, Ph-5-1,* and *Ph-5-2*). These are all identified in wild tomatoes. Three of these genes/genomic regions have been or are used commercially (*Ph-1, Ph-2, Ph-3*), and three others have been reported (*Ph-4, Ph-5-1* and *Ph-5-2*) but are not yet used. However, *P. infestans* is highly diverse and virulent, and isolates that have overcome the *Ph-1, Ph-2, Ph-3* and *Ph-4* resistance genes have already been identified [9]. For the time being (June 2022), the *Ph-5-1* and *Ph-5-2* are considered the most effective resistance genes/genomic regions against tomato late blight, and they might be more effective when combined with the *Ph-3* gene using smart breeding programs and gene pyramiding strategy [10]. Notably, *S. habrochaites* accessions (including LA1775, LA1777, LA2409, LA2099, LA2869 LA1352, LA2855, LA1347, LA1718, and LA1295) offer high levels of resistance to virulence isolates that already overcome *Ph-1*, *Ph-2*, *Ph-3,* and *Ph-4* [11,12]. Several QTLs associated with late blight resistance were mapped on the *S. habrochaites* genome and, out of which, three QTLs were identified on chromosomes 4, 5, and 11, respectively, and fine-mapped using near-isogenic lines [13]. Additionally, Li et al. [11] detected five QTLs in *S. habrochaites* (LA1777) that conferred high levels of resistance to *P. infestans*. Another QTL was mapped on chromosome 11 within a 9.4-cM genomic region of *S. habrochaites* [14]. 

Most recently, two QTLs were detected on chromosomes 6 and 12 of *S. habrochaites*, which are associated with *P. infestans* resistance [15,16]. Furthermore, new metabolomic approaches have enhanced the detection of biochemical markers associated with pest resistance. Trichome secretions play a key role in tomato wild relatives for resistance to different types of insect pests. Six QTLs were detected on chromosomes 1, 7, 8, and 11 of *S. habrochaites* LA1777 that were associated with glandular trichomes type VI and their role in pest resistance [17]. Subsequently, *S. habrochaites* possesses several shapes of trichomes that produce a high number of metabolites released as defence compounds upon stress [18]. 

Plant metabolites are organic compounds generated as an interaction between genetic materials and environmental conditions [19,20]. Exposing plants to various types of stress leads to an array of secondary metabolites, which might have fundamental role in defence mechanisms against diseases and insect pests [21,22,23]. The secondary metabolites can serve as antibiotics and thus influence plant disease and it has been known that leaf extracts can possess antifungal activity. 

Antimicrobial resistance (AMR) or fungicides resistance is a global threat to food production and security. The emergence of fungicide-resistant pathogens can be controlled by regulating genomic and biochemical pathways networks. Host-generated secondary metabolites from the cell wall play a pivotal role in signaling pathways regulation like MAP kinase cascades and cAMP [24]. Therefore, identifying new metabolites associated with fungal disease resistance is indispensable for integrated disease management and to achieve sustainable development goals. 

Remarkably, alternative oxidase (AOX) and transposon-associated sequences, as well as genome-wide association studies, introduce new insights toward the evolution of AM and fungicides resistance. Abu-Nada et al. [25] identified 95 metabolite compounds, of which 42 were associated with pathogenesis-related (PR) metabolites for *P. infestans* in the potato. Volatile organic compounds (VOCs) are organic chemical compounds with a composition that can evaporate under normal temperature and pressure conditions, and such compounds could be used as diagnostic markers. Laothawornkitkul et al. [26] highlighted three volatile metabolites [benzene-ethanol, 5-ethyl-2(5H)-furanone, and (E)-2-hexenal] in potato plants infected with *P. infestans*. The peak area of these compounds was higher in inoculated plants than uninoculated (control). 

In the tomato plant, it is known that leaf extracts from wild tomato plants may have antifungal activity, but to our knowledge, no studies have been conducted to examine such metabolites in detail and the effect on the resistance mechanism of late blight. This study aims to investigate secondary metabolites related to late blight resistance by examining leaf extracts from resistant and susceptible wild tomato accessions. We focus on VOCs, but other compounds may also be responsible. We also aimed to show that the foliar application of leaf extracts from resistant wild tomatoes can potentially be used in biochemical control of late blight in cultivated tomatoes.

## 2. Results

### 2.1. Late Blight Severity on Resistant and Susceptible Tomato Genotypes

Three resistant wild genotypes (*Solanum habrochaites* accession number LA1777, LA2855 and LA1352) and two highly susceptible genotypes (*S. lycopersicum* ‘Super Strain B’ and *S. pimpinellifolium* accession number LA0375) were compared in the current study and differences in the metabolic profiling in plants before and after inoculation with *P. infestans* isolate EG_7 (sub-clonal lineage 23_A1_12) were investigated (Table 1).Disease severity of late blight on the selected accessions was assessed during two growing seasons (2020 and 2021) by applying a whole plant assays carried out under controlled conditions (Table 1 and Table 2). Eight days after inoculation, no symptoms of late blight were observed in the control group (non-inoculated plants). For the plants inoculated with late blight, the statistical analysis (ANOVA) demonstrated highly significant differences among genotypes (*p* = 0.05). The obtained results showed the same pattern in both years (2020 and 2021). No accession was totally immune to the given *P. infestans* isolate, but the most susceptible accessions had almost 100% blight severity (dead). For example, in the 2020 season, the mean disease severity ranged from 2.3 to 99.3% among the accessions (Table 1). The *S. habrochaites* accession LA1777 recorded the highest level of efficacy with 97.7%, followed by the accessions LA2855 and LA1352 (with 91.0 and 90.6% efficacy, respectively). The accessions were categorized in three disease categories: where LA1777 was highly resistant (HR), LA2855 and LA1352 resistant (R), whereas *S. pimpinellifolium* (LA0375) and ‘Super Strain B’ were highly susceptible (HS) (Table 1).

### 2.2. Microscopic Investigation of the Infection

In the resistant accession LA1777, the sporangia germination, appressorium formation, and the mesophyll cells penetration were very slow or absent at 72 HAI (Figure 1A–C). In contrast, all sporangia of *P. infestans* were germinated, and mycelium was colonized and spread in the leaf tissues of the susceptible cultivar ‘Casltlerock’ (Figure 1D). 

### 2.3. Metabolic Profiling in Late Blight Resistant and Susceptible Tomato Genotypes

Our data showed that 31 volatile organic compounds (VOCs) were detected in the five tomato genotypes (Table 3). In the control group (non-inoculated plants), the number of VOCs ranged from 16 to 21 in the three resistant genotypes (LA1352, LA2855, and LA1777), while the numbers were 13 and 14 in the two susceptible genotypes (‘Super Strain B’ and *S. pimpinellifolium* LA0375), respectively. In the inoculated group, the number of VOCs ranged from 11 to 22 in the resistant genotypes while it was 14 and 15 in the susceptible genotypes (*S. pimpinellifolium* LA0375 and ‘Super Strain B’), respectively.

Interestingly, 14 VOCs were detected in resistant genotypes only (the *S. habrochaites* accessions, LA1352, LA2855, and LA1777, and were absent in the susceptible accessions. (Figure 2). These compounds had a retention time from 10.8 to 22.77 min and included p-Menthane-1,2,3-triol, Quercetin 3′,4′,7-trimethyl ether, 7,8-Dihydro-α-ionone, Caryophyllene, β-caryophyllene, geranyl-α-terpinene, 9-cis-Retinoic acid, Ledene oxide, Isocurcumenol, α-Himachalene, 3-Hydroxy-4-methoxybenzaldehyde, Cannabinol, Phytol, and Glycitein. Nine out of 14 VOCs: Quercetin 3′,4′,7-trimethyl ether, 7,8-Dihydro-α-ionone, Caryophyllene, β-Caryophyllen, geranyl-α-terpinene, 9-cis-Retinoic acid, Isocurcumenol, α-Himachalene, Cannabinol, and Glycitein were detected only in LA1777, the most resistant genotype, while p-Menthane-1, 2, 3-triol was detected only in LA1352. It is noteworthy that α-Himachalene in the non-inoculated LA1777 plants was 74-fold higher than in the inoculated LA1777 plants (Figure 2). Two metabolites, Santalol, cis, α- and 3-Octadecenoic acid methyl ester, were detected only in the susceptible accession *S. pimpinellifolium* LA 0375 in the non-inoculated and inoculated plants, respectively (Figure 3).

Furthermore, 6 out of the 31 VOCs included Flavone, 3,4′,5,7-tetramethoxy-, Scopoletin, Nabilone, (+)-α-Tocopherol, Butylated hydroxytoluene, and Carmine acid, and were detected in all non-inoculated and inoculated plants from the five tomato genotypes, and with a retention time (RT) that ranged from 11.18 to 19.90 min (Figure 4; Table 3). Interestingly, peak area percent for these compounds was higher in inoculated plants than in non-inoculated plants, except for Butylated hydroxytoluene and Carmine acid in susceptible tomato cultivar ‘Super Strain B’. Additionally, the lowest peak area (%) of these compounds was represented by LA1777, except for (+)-α-Tocopherol, which recorded the highest peak area in non-inoculated and inoculated plants of LA1777 (Figure 4). In the same context, the highest peak area of the majority of these VOCs was detected in ‘Super Strain B’ and in LA1352.

Additionally, nine compounds were detected in selected genotypes with fluctuation of peak area percentage between inoculated and non-inoculated as well as resistant and susceptible (Figure 4). It is worth mentioning that the peak area (%) of α-Terpinolen and β-Phellandrene was dramatically decreased in inoculated plants compared to non-inoculated plants in resistant accession LA2855, whereas Pentadecanoic acid increased in the same wild species with a retention time 22.147 min (Table 3; Figure 4). In addition, Mesaconic acid and Elaidic acid recorded higher peak areas in susceptible non-inoculated plants (LA0375 and Super Strain B) compared to susceptible, infected plants. Six compounds, p-Menthane-1,2,3-triol, Mesaconic acid, Longipinocarveol, trans-, 3-Hydroxy-4-methoxybenz-aldehyde, 3-Ethyl-5-(2′-ethylbutyl)octadecane, and Elaidic acid, were detected in non-inoculated LA1352 plants but were absent in inoculated LA1352 plants.

### 2.4. Inhibition of VOCs on Mycelial Growth of P. infestans

Figure 5 shows the inhibition of mycelial growth in the presence of VOCs. These VOCs were isolated from the wild tomato species *Solanum habrochaites* accession LA1777. The inhibition of mycelial growth of *P. infestans* isolates reached 100% (completely inhibiting the growth) in the concentrations of 2 mL/L compared to the control without the antagonists.

### 2.5. Efficacy of VOCs on Tomato Late Blight Management

The effect of VOCs application against late blight disease of tomato was assessed using detached leaflet assay under controlled conditions, and the result is shown in Figure 6. The utilized VOCs in the experiment were obtained from the wild-type *Solanum habrochaites* accession LA1777. The obtained result indicates that the targeted treatment dramatically (completely) reduced the infection of late blight. Concerning the whole-plant assay, the effect on late blight of tomato was investigated by using VOCs under controlled conditions. Data in Figure 7 reveals a clear reduction of disease severity by VOCs application compared to the control. On the third day after inoculation, the disease severity of late blight was 3.7% with the spraying of VOCs compared to 23.3% for the control (only *P. infestans*). Furthermore, one week after inoculation, the application of VOCs still had a lower disease severity (22.3%) compared to the control (94.7%) (Figure 7).

## 3. Discussion

Late blight is a major disease in tomato and potato plants, causing acute socioeconomic losses in different regions globally [27,28,29]. Late blight-host resistance, especially quantitative resistance, could decrease plant pathogen progress and develop durable resistance [30]. No late blight-resistant varieties have been released to farmers in Egypt yet. Characterization of late blight resistance mechanisms plays a key role in the success of disease management strategy. However, limited studies are available in this domain, especially in tomato. In a previous study, *S. habrochaites* accessions LA1777, LA1352, and LA2855 have offered high levels of late blight resistance to a virulence isolate EG_7 that has overcome the resistance genes *Ph-2* and *Ph-3* in Egypt [12]. Based on microscopic observations, sporangia germination, infection, and symptom development on the leaves of these genotypes were very slow or absent. This could be due to the metabolites produced by the leaves of these resistant accessions. In this study, GC-MS analysis enabled us to detect 31 metabolites from the leaf surface of three resistant accessions (LA1777, LA1352, and LA2855) and two susceptible genotypes (LA0375 and ‘Super Strain B’), which could help us to explain the resistance mechanism in *S. habrochaites* at the biochemical level.

These compounds revealed uneven peak areas and retention time among all tested tomato genotypes in inoculated and non-inoculated plants. In general, metabolites analysis demonstrated that many sesquiterpene compounds were generated by resistant wild tomato leaves. Metabolites of *Solanum* species, including *S. habrochaites*, *S. peruvianum*, *S. pimpinellifolium*, *S. pennellii,* and *S. lycopersicum* were analyzed and characterized using UHPLC/MS and two-dimensional NMR spectroscopy where Trimethylmyricetin, sesquiterpene acid, fatty acid, Dehydro diterpene acid, Quercetin 3-O-glucosylglucoside, Sesterterpene malonate, Di-dehydroditerpene acid, Tomatine, and several acyl sugars were detected [31,32].

It is clear that the wild species LA1777 recorded the highest numbers of volatile compounds among all tested genotypes, whether in non-inoculated or inoculated plants. These results were consistent with the nature of this species, where varying types of glandular trichomes produce very high levels of VOCs, which could be effective for controlling a wide range of insect pests [25,26,27,28,29,30,31,32,33,34,35]. In the potato wild-type *S. berthaulltii*, the development, and progression of late blight disease have been suppressed by volatiles emitted from the leaves, where two types of glandular trichomes (types IV and VI) have been found [36].

We should acknowledge that the extract may contain compounds that are not VOC and that these may have effects. We focused on VOCs and found that caryophyllene, β-caryophyllene, isocurcumenol, and ledene oxide were detected only in inoculated resistant genotypes, which could be an indication of the role of these compounds in the defence system against *P. infestans*. Yamagiwa et al. [37] found a positive effect of the volatile compound β-caryophyllene in *Brassica campestris* var. *perviridis* with increased resistance against crucifer anthracnose (*Colletotrichum higginsianum*). Moreover, ample of the literature mention that the aforementioned compounds possess biological activities such as insecticidal, antifeedant, antifungal, and antibacterial effects [38,39,40]. It is proposed that the mode of action of β-caryophyllene is by transcriptional activation of some defence-related genes that have antifungal activity [37,41]. Interestingly, VOCs can act as damage-associated molecular patterns (DAMPs) with local and systemic responses in plants. Moreover, plant pathogens attack can induce the emission of VOC that are associated with the defence system. 

Geranyl-α-terpinene and 9-cis-retinoic acid were detected only in the highly resistant wild species LA1777 with an increase in plants after exposure to spore suspension; therefore, emitting of these compounds might be related to disease resistance in biological systems. Naka et al. [42] mentioned that 9-cis-retinoic acid plays a crucial role as a drug in cancer cell treatment and inhibition of its proliferation. In addition, mesaconic acid and longipinocarveol, trans-, demonstrated remarkably reduced susceptibility in the inoculated genotypes compared to the non-inoculated, with an increase in some resistant infected genotypes, which probably play a concrete role in disease resistance. Furthermore, α-himachalene and cannabinol were detected only in inoculated and non-inoculated plants of LA1777. These compounds have been shown to have antibacterial and antifungal, activities in medicinal plants cinnamon, anise, black seed and clove against pea root-rot fungus *Rhizoctonia solani* [43,44]. 

Interestingly, glycitein was detected only in inoculated plants of LA1777, which might have a role in the defence reaction as a biochemical or metabolic pathway in plants against late blight. Glycitein is one of the most important bioactive isoflavones that can be generated in plant species post-treatment with biotic elicitors such as *Aspergillus niger* and *Rhizopus oligosporus* [45]. Several studies suggested that isoflavonoids are indicator molecules for the synthesis of antimicrobial substances (phytoalexins) in plant cells through host-pathogen or host-insect responses as a part of the plant defence system [46,47]. In addition, the flavonoid revealed antifungal activities against soil-borne fungi like *Fusarium oxysporum* f. sp. *lycopersici* [48]. Therefore, isoflavonoids have been deeply nested with plant immune systems to restrain plant pathogens [49]. 

The peak areas of α-terpinolen and β-phellandrene were dramatically decreased in inoculated resistant plants of LA2855 compared with non-inoculated plants. These compounds were also detected in inoculated ‘Super Strain B’, which may be evidence for the lack of role of these compounds in the defence response. Our results are in agreement with Holopainen et al. [50], wherein the emissions of some monoterpenes like β-phellandrene quantity decreased after feeding by *Hylobius abietis* on *Pinus sylvestris* seedlings. 

We detected some VOCs as diagnostic markers, such as 3-Octadecenoic acid, methyl ester, and glycitein, for late blight infection as well as p-Menthane-1,2,3-triol and Santalol, cis,α- for healthy samples. However, further chemical analysis using LC-MS and GC-MS in F2 populations derived from LA1777 is underway to confirm the results obtained from this study. In a previous study on potato, benzene-ethanol and (E)-2-hexenal,5-ethyl-2(5H)-furanone were detected as markers for *P. infestans* infection [26]. Intriguingly, several studies mention that VOCs emitted by fungi or plant species have the potential for systemic induced resistance through different metabolic pathways and biocontrol agents against an array of destructive plant pathogens [51,52,53,54,55]. Application of leaf extracts or VOCs found in resistant wild tomato genotypes could serve as an eco-friendly and alternative method to spraying. Recently Kong et al. [56] mentioned that VOCs inhibited the mycelial growth of *Colletotrichum gloeosporioides*, the pathogen of *Liriodendron chinense* × *Tulipifera* black spot in vitro, reaching 63%. The fungal cell membrane is considered the first defence line to protect the cell and other normal components under stress conditions. When this membrane is devastated, some intracellular compounds like, phosphates, proteins, carbonates, and nuclear acids (DNA and RNA) will be unrestricted, subsequently leading to cell death. Additionally, malondialdehyde (MDA) is one of the most significant products that can be used as an indication of the damage level of the membrane system [57]. Likewise, Kong et al. [56] reported that the content of MDA in the mycelium of *C. gloeosporioides*, was increased in the treated plates with VOCs compared to the control plates in vitro assay. Therefore, it is projected that the VOCs can act against *P. infestans* in vitro by different approaches, such as direct influence as a poisons material and/or increasing the level of MDA in the mycelium, as well as destroying the permeability of cell membrane and thereafter having an effect on the integrity of the cell, resulting in mycelial growth inhibition.

In our future study, it will be important to test metabolites identified in this study singly or in combinations against late blight and other tomato serious pathogens and to test direct VOCs-pathogen interactions. Furthermore, detected compounds can be utilized as selective markers to distinguish between incompatible/compatible and non-infected/infected genotypes. This could be useful knowledge for late blight resistance breeding.

## 4. Materials and Methods

### 4.1. Plant Material and Condition

Five tomato genotypes were selected based on previous late blight resistance screening [12]. These included three resistant accessions of the wild species *Solanum habrochaites* (LA 1352, LA 2855, and LA 1777) and two susceptible accessions, one belonging to the wild species *S. pimpinellifolium* (LA 0375) and one *S. lycopersicum* commercial cultivar ‘Super Strain B’ (Table 1). The screenings were done using the *Phytophthora infestans* isolate EG_7 [12].

The workflow of the current study is presented in Figure 8. The tomato wild relative accessions (wild genotypes) were requested from C. M. Rick Tomato Genetics Resource Center (TGRC), University of California (Davis, CA, USA). These have LA accession numbers. The commercial tomato cultivar was obtained from Horticulture Research Institute, Agricultural Research Center (ARC, Giza, Egypt). The cultivation was done in the greenhouse (25 ± 2 °C, 16/8 h day/night), seeded in 209-cell trays containing 40 mL per cell filled with a growth medium of peat moss-vermiculite mixture (1:1 volume) (Al Kalthoum Agricultural Co., Mansoura, Egypt). Plants were watered daily and fertilized weekly with NPK 15-15-15 (Al Kalthoum Agricultural Co., Mansoura, Egypt). Four weeks after sowing, seedlings were transplanted into 20 cm pots containing potting soil. Eight-week-old plants, on average 30 cm high, were moved from the greenhouse to a growth room (20  ±  2 °C, 90 % relative humidity (RH), 16/8 h day/night) at the Plant Pathology Research Institute, ARC (Egypt) for *P. infestans* inoculation as described by Arafa et al. [12].

### 4.2. Phytophthora infestans Inoculum

*Phytophthora infestans*, isolate EG_7, was requested from the Plant Pathology Research Institute, ARC (Egypt). This isolate was originally collected from tomatoes in the Kafrelsheikh governorate (Egypt) and was identified at James Hutton Institute (Dundee, UK) based on 12 simple sequence repeat markers [58,59]. The EG_7 isolate was cultured and maintained on a rye-sucrose agar (RSA) at 18 °C [60], and antibiotics were added to the medium as described by Arafa et al. [59]. 

The inoculum of *P. infestans* was prepared as described by Arafa et al. [12]. Briefly, the leaves of susceptible tomato variety were placed on moistened paper in Petri plates and inoculated with 30 µL of sporangial suspension/leaflet to propagate the sporangia. The Leaflets were incubated for 10 days at 18 °C, placed in a 500 mL glass beaker with water, and gently shaken to dislodge sporangia. The concentration of the sporangia was examined by a haemocytometer and adjusted to 15 × 10^4^ sporangia/mL. Prior to inoculation, the suspension was chilled at 4 °C for 2–4 h to encourage zoospore release [61].

### 4.3. Inoculation and Assessment of Late Blight Disease 

The plants from each of the five genotypes were divided into two: one group to be inoculated and one group not to be inoculated, and with three replicate plants per accession and treatment. The inoculated group was sprayed with the sporangial suspension using a hand sprayer until complete leaf coverage under controlled conditions at 20/18 °C day/night and 95 ± 2 % RH with a 12 h photoperiod at a growth chamber. The other group was sprayed with sterilized distilled water as a control. The disease severity was performed on the whole-plant assay using a scale of 0 to 6 [5,8]. Infection was assessed on day 0, 5, 10, 15, and 20 post-inoculation. The calculation of disease severity was done according to the equation described by Descalzo et al. [62]: R = [∑ (a × b)/n × K] × 100
where R is the disease severity (in %), a is the number of infected leaves rated, b is the score in a numerical value of each grade, n is the total number of examined plants, and K is the highest degree of infection in the scale. 

The disease severity (in %) after reaching the complete damage of the check cultivar (highly susceptible) was calculated. Additionally, the area under the disease progress curve (AUDPC) was determined for each tomato genotype [63] during two growing seasons, 2020 and 2021, as follows: AUDPC = D [1/2 (Y1 + YK) +Y2 + Y3 +.... + Y (K − 1)]
where D is the days between each successive two readings, Y1 is the first disease record, and YK is the last disease record.

The efficacy of the tested tomato genotypes was determined according to the following formula: $$\mathrm{Efficacy}\%=\frac{\mathrm{DS}\%\;\mathrm{control}\;-\;\mathrm{DS}\%\;\mathrm{treatment}\;}{\mathrm{DS}\;\mathrm{control}}\times 100$$

### 4.4. Microscopic Investigation of the Infection

Six tomato leaflets for each tested genotype were examined by light microscope [64] with a Leica DM1000 (Leica Microsystems, Wetzlar, Germany) at the Plant Pathology and Biotechnology Laboratory (PPBL), Faculty of Agriculture, Kafrelsheikh University (Egypt), in order to observe sporangia germination and spread of *P. infestans* mycelium in leaf tissues. The inoculated areas with sporangial suspension were stained and cleared using the Trypan Blue Protocol [65] to observe *P. infestans* structures inside host cells. At 72 h after inoculation (HAI), tomato leaflets were fixed on wipes or Whatman filter paper (Fisher Scientific, Pittsburgh, PA, USA) into petri dishes moistened with glacial acetic acid: ethanol (1:3 *v*/*v*) for two days. Then, the samples were transferred to new petri dishes moistened with distilled water for 5 h and moved the samples to other petri dishes containing wipes moistened with lactic acid: glycerol: distilled water (1:1:1) [66] up to microscopic examination.

### 4.5. Gas Chromatography/Mass Spectrometry (GC/MS) Analysis of Volatiles

The analyses were conducted at the Mass Spectrometry Laboratory (MSL) at The Regional Center for Food and Feed (RCFF, ISO/IEC 17025:2005), ARC, Egypt. Volatile organic compounds (VOCs) analysis was done on all five genotypes, and with three replicate analysis per genotype and treatment. For each genotype, leaflets of the third or fourth leaves from the apex were sampled eight weeks after sowing and three days after inoculation, and 1 g was used for each sample for volatiles identification. Before analysis, each leaf sample was washed in 30 mL of methylene chloride in a 40 mL vial for 3 h at 200 rpm using an Innova 2100 platform shaker (New Brunswick Science, Edison, NJ, USA). The solvent rinse (with the trichomes and leaf surface contents) was decanted using a Whatman no. 4 filter paper (Fisher Scientific, Pittsburgh, PA, USA), and the filtrate, 1 μL of each sample, was injected into the GC-MS instrument with a GC (Agilent Technologies 7890A, Santa Clara, CA, USA) interfaced with a mass selective detector, MSD, Agilent 7000 Triple Quad (Santa Clara, USA) equipped with an apolar Agilent HP-5ms, which is a (5%-phenyl)-methyl poly siloxane capillary column, 30 × 0.25 mm i. d. and 0.25 μm in film thickness. The carrier gas was helium with a linear velocity of 1 mL/min. The temperatures at the injector and detector s were 200 °C and 250 °C, respectively. Injection mode was split with a ratio 1:10. Operating parameters included: ionization potential = 70 eV, interface temperature = 250 °C, and acquisition mass range = 50–600. The identification of the components was based on a comparison of their mass spectra and retention time with reference compounds (standards), and a computer matching was done with the libraries at National Institute of Standard and Technology (http://www.nist.gov/, accessed on 12 April 2022) and Wiley (http://www.wileyregistry.com/, accessed on 12 April 2022). 

### 4.6. Effect of VOCs on P. infestans under In Vitro Conditions

To assay the inhibitory in vitro effects of VOCs on the mycelial growth of the *P. infestans* isolate, the VOCs were extracted from resistant tomato wild relatives *S. habrochaites* LA1777 as described above. Seeds of the wild tomato relatives, *S. habrochaites* LA1777, were sown under controlled conditions (26 ± 2 °C, 70 % humidity, 16/8 h day/night) for three months. During the vegetative growth period, all agricultural practices, including irrigation and fertilization, were applied according to the recommendations of the horticultural Research Institute, ARC (Egypt). Six weeks after transplanting, fresh leaves of LA1777 were collected and rinsed in methylene chloride, then filtered as aforementioned (Section 4.4). The solution was sterilized using Sterile PTFE Syringe Filters, pore size 0.45 um (Sigma Aldrich, Darmstadt, Germany). The sterilized solution was amended with melted rye agar medium at 45 °C. The medium was poured into plates (90 mm diameter) and inoculated with 3 mm mycelium disk (two weeks old pathogen cultures) of *P. infestans* at the center of the plate. The petri dish cultures were incubated at 19 ± 2 °C in the dark, and the mycelial growth was measured every five days after incubation. The rye agar plates without VOCs were considered the control. The current experiment was run twice with five replicates per run. To assess the effect of treatment, the colony diameter of *P. infestans* mycelial growth in the control and in the treated Petri dishes was measured and the effect was calculated by the following formula:Mycelial growth inhibition (%) = [(dc − dt)/dc] × 100
where dc is the average diameter of *P. infestans* mycelial growth in control Petri dishes, and dt is the average diameter of *P. infestans* mycelial growth in the treated Petri dishes.

### 4.7. Effect of VOCs on P. infestans in Leaflet and Whole Plant Assays

The VOCs were applied in trials based on two approaches: a detached leaflet assay and a whole-plant assay, respectively. The spore suspension of *P. infestans* was prepared as aforementioned with the adjusted concentration. For the detached leaflet assay, the leaves of seven-week-old tomato plants of a susceptible commercial cultivar ‘Castlerock’ were dipped in the suspension of VOCs for 20 s. and subsequently inoculated with *P. infestans* suspension in dishes 9 cm in diameter. The plates were incubated at 19 ± 2 °C, and disease severity was assessed five and seven days post-inoculation. Concerning the whole-plant assay, the VOC suspension was sprayed on the susceptible tomato leaves (cv. ‘Castlerock’) after 48 h of pathogen inoculation in a growth chamber with the same incubation conditions as already described. The disease severity was assessed three, five, and seven days after inoculation. The positive control included tomato plants infected with *P. infestans*.

### 4.8. Statistical Analysis

Analysis of variance (ANOVA) was conducted with complete randomized block design models, using WASP software (Web Agriculture Stat Package, https://ccari.icar.gov.in/waspnew.html, accessed on 23 April 2022). The comparisons of means were determined by Duncan’s multiple range tests at a 95% significance level (*p* ≤ 0.05) [67]. The least significant difference tests (LSD) were used to compare mean differences between genotypes over all the studied environments [68].

## 5. Conclusions

It could be concluded that 31 VOCs were identified in this study. Tomato wild accession LA 1777 recorded the highest numbers of VOCs. Furthermore, the application of VOCs in whole plant assay decreased the late blight disease severity compared to control (untreated plants). Additionally, the in vitro assay showed that the VOCs revealed a complete reduction in the *P. infestans* mycelial growth. Therefore, metabolites in wild late blight-resistant accessions might play a key role in understanding the resistance mechanisms. We linked these metabolites to the glandular trichomes on the leaves of resistant genotypes. Further studies are needed to extract metabolites from glandular trichomes and test them against a diversity of microorganisms and insects. Such investigations can open future perspectives on plant immune and defense systems and the treatment and diagnosis of plant diseases. 

## Figures and Tables

**Figure 1 plants-11-01824-f001:**
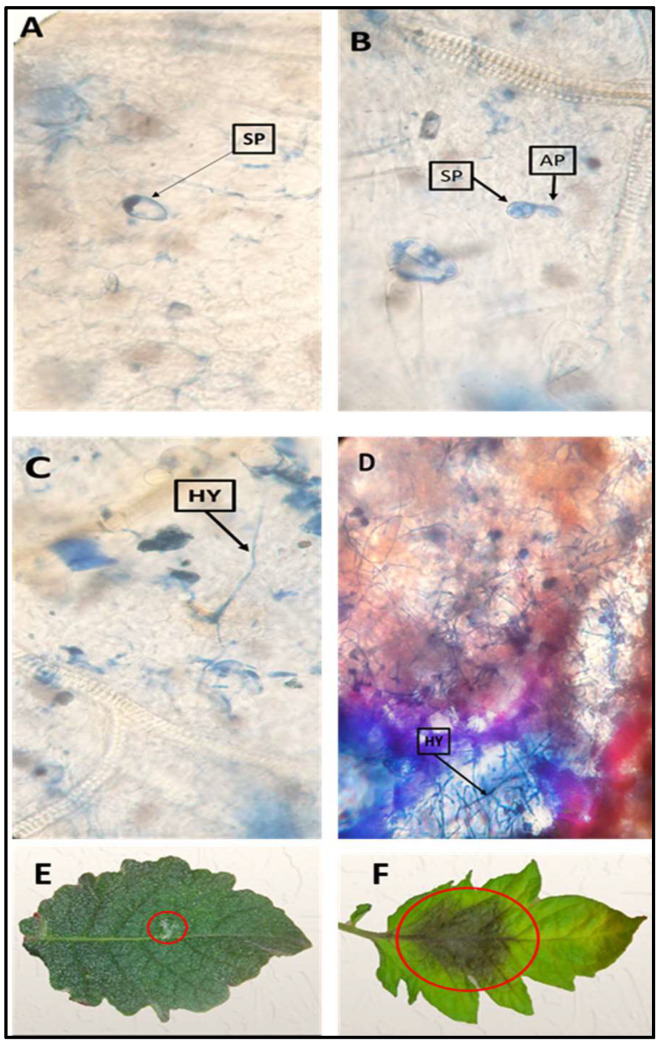
Observations of sporangia germination progression of *Phytophthora infestans* on tomato wild relative leaves under a light microscope at 72 h after inoculation (HAI) (**A**–**D**): (**A**) no germination of sporangium; (**B**,**C**) very slow germination of sporangia, appresorium formation and weak penetration on *S. habrochaites* accession LA1777 leaves as a resistant wild type to late blight; (**D**) colonization of intense mycelium inside host cells and penetration of the leaf tissue on the susceptible tomato cultivar Castlerock. Evaluation of resistance and lesion progression of late blight on tomato leaflets of two germplasm at 96 HAI in vitro; (**E**) LA1777; (**F**) Castlerock cultivar. Sp—Sporangium; Ap—Appressorium; Hy—Hypha.

**Figure 2 plants-11-01824-f002:**
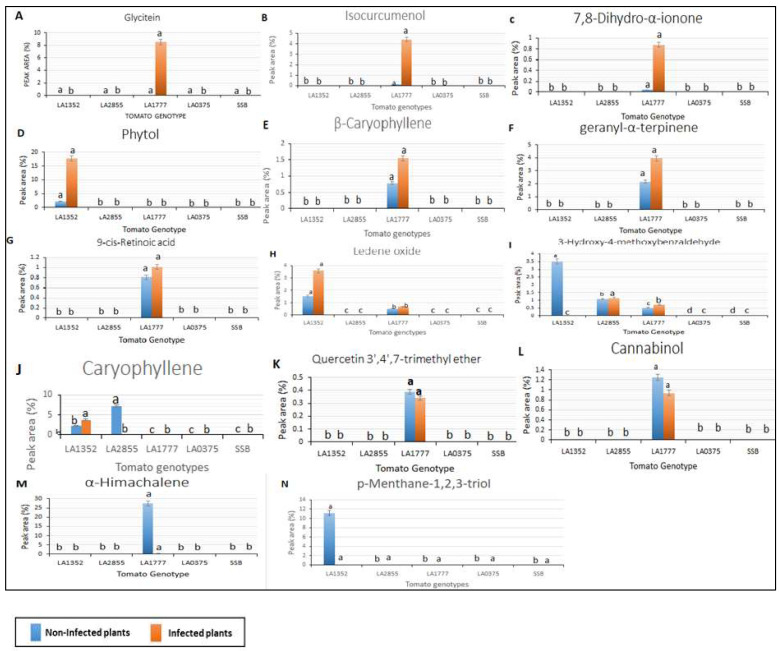
Peak area of volatile organic compounds detected in only resistant tomato wild species using gas chromatography/mass spectrometry. SSB—Super Stain B. The blue bar indicates the control group that was not inoculated (non-infected plants); the orange bar indicates the inoculated group (infected plants). Each compound is presented in a bar chart: (**A**) Glycitein (**B**) Isocurcumenol (**C**) 7,8-Dihydro-α-ionone (**D**) Phytol (**E**) β-Caryophyllene (**F**) geranyl-α-terpinene (**G**) 9-cis-Retinoic acid (**H**) Ledene oxide (**I**) 3-Hydroxy-4-methoxybenzaldehyde (**J**) Caryophyllene (**K**) Quercetin 3′,4′,7-trimethyl ether (**L**) Cannabinol (**M**) α-Himachalene (**N**) p-Menthane-1,2,3-triol. Results are given as mean values ± standard deviations. Values in the same bar chart with different letters are significantly different at *p* < 0.05.

**Figure 3 plants-11-01824-f003:**
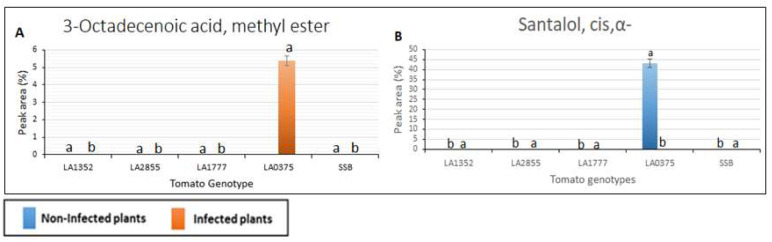
Volatile organic compounds detected only in susceptible tomato genotypes using gas chromatography/mass spectrometry. SSB—Super Stain B. The blue bar indicates the control group that was not inoculated (non-infected plants); the orange bar indicates the inoculated group (infected plants). Results are given as mean values ± standard deviations. Each compound is presented in a bar chart: (**A**) 3-Octadecenoic acid, methyl ester (**B**) Santalol, cis, α-. Values in the same bar chart with different letters are significantly different at *p* < 0.05.

**Figure 4 plants-11-01824-f004:**
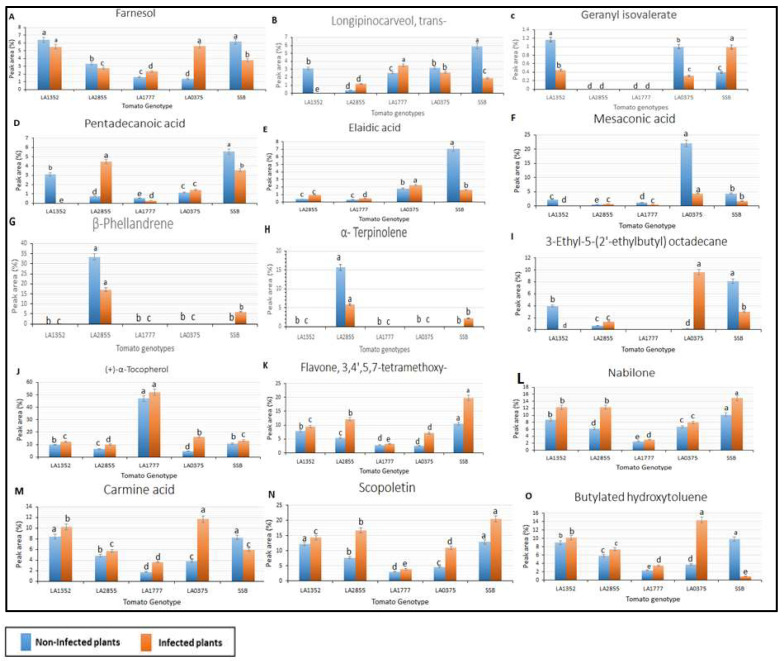
Volatile organic compounds were detected in both resistant and susceptible tested tomato genotypes with fluctuation of peak area percentages of infected and non-infected plants using Gas chromatography/mass spectrometry. SSB—Super Stain B. The blue bar indicates the control group that was not inoculated (non-infected plants); the orange bar indicates the inoculated group (infected plants). Each compound is presented in a bar chart: (**A**) Farnesol (**B**) Longipinocarveol, trans- (**C**) Geranyl isovalerate (**D**) Pentadecanoic acid (**E**) Elaidic acid (**F**) Mesaconic acid (**G**) β-Phellandrene (**H**) α-Terpinolene (**I**) 3-Ethyl-5-(2′-ethylbutyl) octadecane (**J**) (+)-α-Tocopherol (**K**) Flavone, 3,4′,5,7-tetramethoxy- (**L**) Nabilone (**M**) Carmine acid (**N**) Scopoletin (**O**) Butylated hydroxytoluene. Results are given as mean values ± standard deviations. Values in the same bar chart with different letters are significantly different at *p* < 0.05.

**Figure 5 plants-11-01824-f005:**
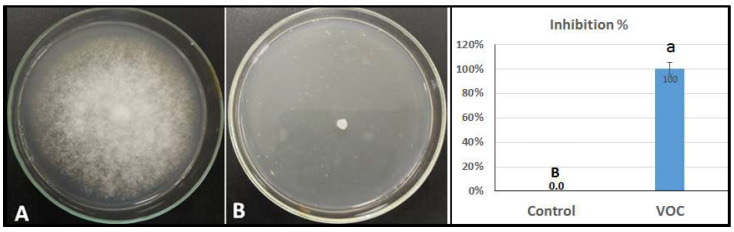
Effect of the extracted volatile organic compounds (VOCs) on the mycelial growth of the test *P. infestans* isolate. (**A**) Control plate of *P. infestans* (normal growth); (**B**) VOCs amended (no *P. infestans* growth). Values in the same bar chart with different letters are significantly different at *p* < 0.05.

**Figure 6 plants-11-01824-f006:**
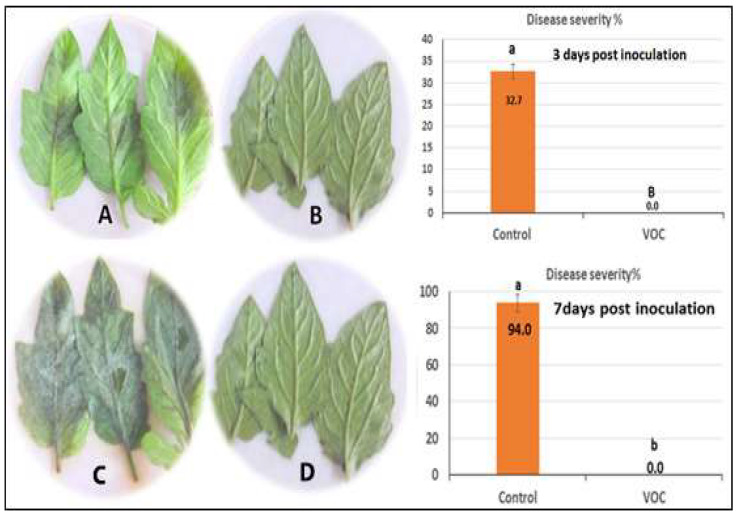
Efficacy of VOCs on tomato late blight using detached leaflet assay. (**A**) Positive control; (**B**) suppression of *P. infestans* of tomato leaves where there is no infection or any symptoms on susceptible variety; (**C**) Positive control of late blight disease seven days post-inoculation; (**D**) disease severity percentage of late blight seven days post-inoculation where there is no infection of *P. infestans*. Values in the same bar chart followed by different letters are significantly different at *p* < 0.05.

**Figure 7 plants-11-01824-f007:**
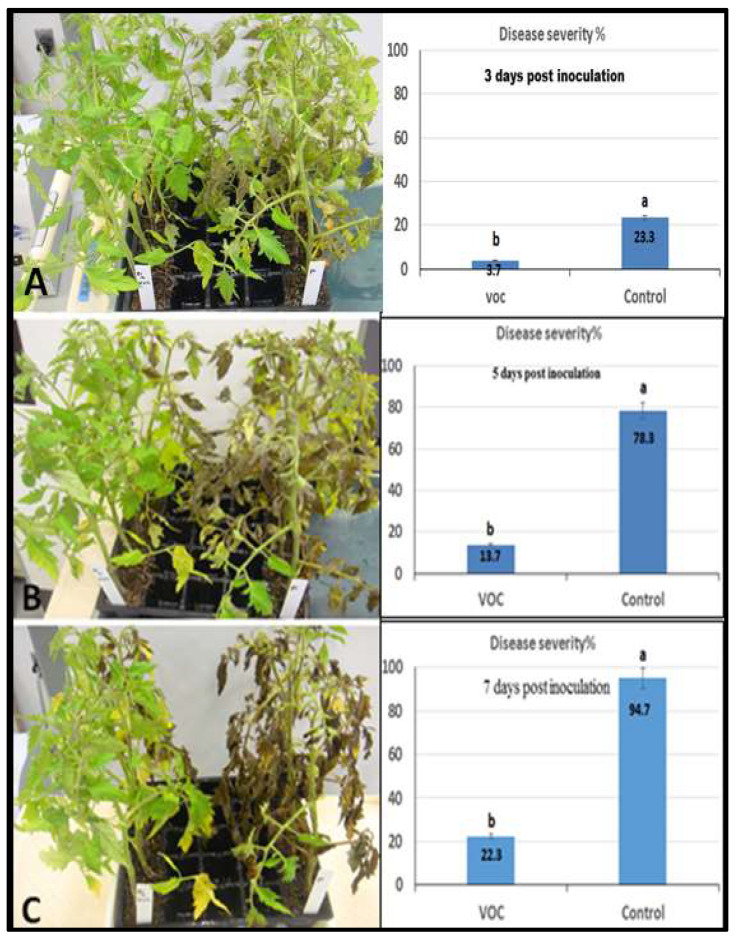
Efficacy of VOCs on tomato late blight through a whole-plant approach where the left row refers to sprayed plants by *P. infestans* and VOCs whereas the right row refers to infected plants (positive control). (**A**) Symptoms three days post-inoculation, (**B**) symptoms five days post-inoculation, (**C**) symptoms seven days post-inoculation. Values in the same bar chart with different letters are significantly different at *p* < 0.05.

**Figure 8 plants-11-01824-f008:**
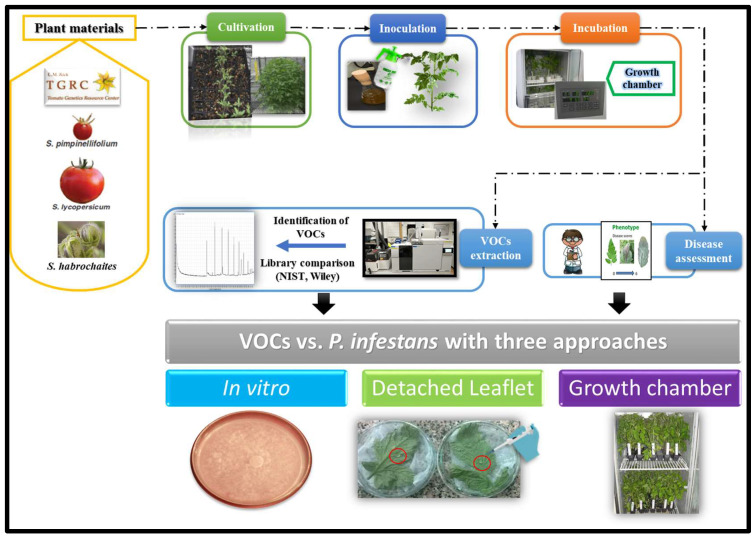
Workflow of volatile organic compounds emission in tomato wild relatives and their extraction using gas chromatography-mass spectrometry as well as their impact on tomato late blight caused by *P. infestans*.

**Table 1 plants-11-01824-t001:** Late blight disease severity percentage for four wild tomato accessions and commercial variety ‘Super Strain B’ inoculated with *P. infestans* isolate EG_7 under greenhouse conditions. Results from 2020 ^1^.

S No.	Taxon/Accession	Disease Severity (%)					Efficacy (%)	AUDPC	Disease Response
		Zero Time	5 DPI	10 DPI	15 DPI	20 DPI			
1	*S. habrochaites* (LA1777)	0.0	0.7	1.4	1.9	2.3 d	97.7	25.8 e	HR
2	*S. habrochaites* (LA2855)	0.0	0.8	1.9	3.4	8.9 c	91.0	52.8 d	R
3	*S. habrochaites* (LA1352)	0.0	1.3	2.4	4.1	9.3 c	90.6	62.3 c	R
4	*S. pimpinellifolium* (LA0375)	0.0	24.3	63.7	77.3	93.7 b	5.63	1060.8 b	HS
5	‘Super Strain B’ (Cultivar)	0.0	27.1	62.3	87.4	99.3 a	0.0	1132.3 a	HS
L.S.D at 0.05						2.38		9.03	

^1^ Values in the same column followed by different letters are significantly different at *p* < 0.05. DPI—Days Post-Inoculation; AUDPC—Area Under the Disease Progress Curve; HR—Highly Resistant; R—Resistant; HS—Highly Susceptible.

**Table 2 plants-11-01824-t002:** Disease severity of late blight for five tomato genotypes inoculated with *P. infestans* isolate EG_7 under controlled conditions. Results from 2021 ^1^.

S. No.	Taxon/Accession	Disease Severity (%)					Efficacy (%)	AUDPC	Disease Response
		Zero Time	5 DPI	10 DPI	15 DPI	20 DPI			
1	*S. habrochaites* (LA1777)	0.0	1.1	1.9	2.3	3.3 e	96.7	34.75 d	HR
2	*S. habrochaites* (LA2855)	0.0	1.8	2.7	3.9	9.9 d	90.1	66.75 c	R
3	*S. habrochaites* (LA1352)	0.0	2.0	2.9	4.8	10.3 c	89.7	74.25 c	R
4	*S. pimpinellifolium* (LA0375)	0.0	28.3	64.8	79.7	95.3 b	4.7	1102.3 b	HS
5	‘Super Strain B’ (Commercial cultivar)	0.0	37.1	67.9	88.4	100 a	0.0	1217.0 a	HS
	L.S.D at 0.05					2.973		10.418	

^1^ Values in the same column followed by different letters are significantly different at *p* < 0.05. DPI—Days Post-Inoculation; AUDPC—Area Under the Disease Progress Curve; HR—Highly Resistant; R—Resistant; HS—Highly Susceptible.

**Table 3 plants-11-01824-t003:** Volatile organic compounds identified in the leaf extract of tomato genotypes using GC/MS and where RT—Retention Time, MF—Molecular Formula, and MW—Molecular Weight (in g/mol).

Name	RT	MF/MW
α-Terpinolene	6.63	C_10_H_16_/136.238
β-Phellandrene	7.01	C_10_H_16_/136.238
p-Menthane-1,2,3-triol	10.80	C_10_H_20_/188.264
Flavone, 3,4′,5,7-tetramethoxy-	11.18	C_19_H_18_O_6_/342.347
Quercetin 3′,4′,7-trimethyl ether	11.70	C_18_H_16_O_7_/344.319
7,8-Dihydro-α-ionone	12.60	C_13_H_22_O/194.318
Caryophyllene	12.70	C_15_H_24_/204.357
β-caryophyllene	13.04	C_15_H_24_/204.357
Scopoletin	13.38	C_10_H_8_O_4_/192.17
geranyl-α-terpinene	14.16	C_20_H_32_/272.47
Santalol, cis, α-	15.03	C_15_H_24_O/220.356
9-cis-Retinoic acid	15.22	C_20_H_28_O_2_/300.44
Nabilone	15.33	C_24_H_36_O_3_/372.549
Ledene oxide	15.46	C_15_H_24_O/220.356
Mesaconic acid	15.87	C_5_H_6_O_4_/130.099
Longipinocarveol, trans-	16.32	C_15_H_24_O/220.356
Geranyl isovalerate	16.88	C_15_H_26_O_2_/238.37
(+)-α-Tocopherol	17.01	C_29_H_50_O_2_/430.71
Isocurcumenol	17.50	C_15_H_22_O_2_/234.339
α-Himachalene	17.58	C_15_H_24_/204.357
3-Hydroxy-4-methoxybenzaldehyde	18.02	C_8_H_8_O_3_/152.149
3-Octadecenoic acid, methyl ester	18.03	C_19_H_36_O_2_/296.495
3-Ethyl-5-(2′-ethylbutyl) octadecane	18.45	C_26_H_54_/366.718
Butylated hydroxytoluene	18.52	C_15_H_24_O/220.36
Cannabinol	19.38	C_21_H_26_O_2_/310.437
Carmine acid	19.90	C_22_H_20_O_13_/492.389
Elaidic acid	20.01	C_18_H_34_O_2_/282.468
Phytol	20.50	C_20_H_40_O/296.539
Farnesol	21.17	C_15_H_26_O/222.372
Pentadecanoic acid	22.15	C_15_H_30_O_2_/242.403
Glycitein	22.77	C_16_H_12_O_5_/284.267
α-Terpinolene	6.63	C_10_H_16_/136.238

## Data Availability

The data presented in this study are available on request from the corresponding author.

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
