# Peer review of "Leaf Extracts from Resistant Wild Tomato Can Be Used to Control Late Blight (Phytophthora infestans) in the Cultivated Tomato"

_plants, 2022, doi:10.3390/plants11141824_

Round 1
Reviewer 1 Report
The revised version of this MS is now well written.
Author Response
The revised version of this MS is now well written.
Thank you for taking your time and for the feedback you have provided. We made a few English language edits as suggested.
Reviewer 2 Report
I congratulate the authors for the work and manuscript. Manuscript ID plants-1818911 " Leaf Extracts from Resistant Wild Tomato can be used to Control Late Blight (Phytophthora infestans) in Cultivated Tomato" by Arafa and collaborators describe an investigation of the volatile organic compounds present in resistant and susceptible tomato varieties and their effect on growth of Phytophthora infestans. Although the precise molecular mechanisms underlying the growth inhibition has not been elucidated, the authors point to future studies aiming to clarify this. The work should be interesting to the general readership of Plants and should be ready for publication after minor revisions I listed below.
line 43: review punctuation.
line 45, 60, 88, 296, and 469: remove excess space between words.
line 89: replace "fungicide-resistance" with "fungicide-resistant".
line 101: replace "volatiles" with "volatile".
line 105: replace "to detailed examine such metabolites" with "to examine such metabolites in detail, and"
Figure 6 legend: include description of panels C and D.
The sentence in lines 301-303 should be revised for clarity. It is unclear how PAMPs can be suppressed as they are constituents of the microorganisms.
line 307: replace "play" with "plays".
line 308: replace "inhibition its" with inhibition of its".
line 405: remove double punctuation.
Author Response
I congratulate the authors for the work and manuscript. Manuscript ID plants-1818911 " Leaf Extracts from Resistant Wild Tomato can be used to Control Late Blight (Phytophthora infestans) in Cultivated Tomato" by Arafa and collaborators describe an investigation of the volatile organic compounds present in resistant and susceptible tomato varieties and their effect on growth of Phytophthora infestans. Although the precise molecular mechanisms underlying the growth inhibition has not been elucidated, the authors point to future studies aiming to clarify this. The work should be interesting to the general readership of Plants and should be ready for publication after minor revisions I listed below.
Thank you for taking your time and for the feedback you have provided. Here is our detailed feedback:
line 43: review punctuation. DONE
line 45, 60, 88, 296, and 469: remove excess space between words. DONE
line 89: replace "fungicide-resistance" with "fungicide-resistant". DONE, now in line 87
line 101: replace "volatiles" with "volatile". DONE
line 105: replace "to detailed examine such metabolites" with "to examine such metabolites in detail, and" DONE in line 104
Figure 6 legend: include description of panels C and D.
We have now added to the figure description (now new line 246-247):
;(C) Positive control of late blight disease 7 days post inoculation; (D) disease severity percentage of late blight 7 days post inoculation where there is no infection of P. infestans
The sentence in lines 301-303 should be revised for clarity. It is unclear how PAMPs can be suppressed as they are constituents of the microorganisms.
We replaced the original sentence «Additionally, the pathogen-associated molecular patterns (PAMPs) which are generate from varying types of microorganisms including P. infestans can be suppressed by VOCs.»
with the following new paragraph (now line 298-301): “Interestingly, VOCs can act as damage-associated molecular patterns (DAMPs) with local and systemic responses in plants. Moreover, plant pathogens attack can induce the emission of VOC that are associated with the defence system.”
line 307: replace "play" with "plays". DONE, now line 309
line 308: replace "inhibition its" with inhibition of its". DONE, now line 306
line 405: remove double punctuation. DONE
This manuscript is a resubmission of an earlier submission. The following is a list of the peer review reports and author responses from that submission.
Round 1
Reviewer 1 Report
The manuscript presents some new and interesting data on VOC from resistant and susceptible tomato plants and their interaction with P. infestans. The results increase our understanding of this important plant-pathogen interaction and even might lead to the development of a new control strategy. I recommend the publication of the paper after some modifications and improvements. Several parts of the manuscript, e.g. some details of the methods, are not very clear, and some results should be discussed in more details.
Please improve the following parts:
Line 205 ff and Figure 5: The origin of the VOCs (LA1777) should be added. The very strong inhibitory effect of the VOCs on the growth of P. infestans should be discussed in more details. Most VOCs act by triggering a response in plant, which then might help the plants to survive pathogen or pest attack, and not by direct interaction with the pathogen.
Line 215 ff and Figure 7: The authors should mention, which VOCs were used for the application experiment. Based on Material and Method line 421 I assume that also for this experiment the VOCs from, LA1777 were used, but it is not very clear.
Line 326 and Table 1: The authors describe the use of the commercial strain ‘Super Strain B’ for VOC analysis. However, the efficacy of VOCs was tested on another susceptible cultivar “Castlerock’. The authors should describe why they used two different cultivars for those experiments.
Line 396ff: VOC are metabolites which express their physiological reaction after being emitted from the producing plant. Therefore, most studies on VOCs use air sampled from the immediate vicinity of the plants for collecting VOCs. In the presented study, metabolites were isolated directly from leaflets, which could have an impact on the results and should be explained and discussed.
Line 422: The method is not describe clearly. ‘The prepared solution was amended with rye agar medium 422 and inoculated with 3 mm mycelium disk’. More details needed.
The authors should double-check the manuscript for minor mistakes with the English language and some typos.
Author Response
We have attached our revised manuscript, " Leaf Extracts from Resistant Wild Tomato Can be Used to Control Late Blight (Phytophthora infestans) in Cultivated Tomato”, which is submitted to Plants Journal (plants-1756720).
After careful consideration of the reviewers’ comments, we made a number of major revisions and improvements. The revised manuscript is submitted with using the Track Changes function in Word but we have also attached the same document with all changes accepted so it is easier to read
Attached is a detailed feedback to the comments from you and the two other reviewers. Our response is in bold.
Reviewer 1
The manuscript presents some new and interesting data on VOC from resistant and susceptible tomato plants and their interaction with P. infestans. The results increase our understanding of this important plant-pathogen interaction and even might lead to the development of a new control strategy. I recommend the publication of the paper after some modifications and improvements. Several parts of the manuscript, e.g. some details of the methods, are not very clear, and some results should be discussed in more details.
Thank you for these comments. We have made changes in several parts of the manuscript, for example as recommended in the methods section where it should be clear in all parts now. We also added paragraphs in the discussion section that makes the outcome/value from our research clearer (see below specific replies).
Please improve the following parts:
Line 205 ff and Figure 5: The origin of the VOCs (LA1777) should be added. The very strong inhibitory effect of the VOCs on the growth of P. infestans should be discussed in more details. Most VOCs act by triggering a response in plant, which then might help the plants to survive pathogen or pest attack, and not by direct interaction with the pathogen.
Thanks for the comment. The origin of the VOCs (LA1777) has added to the manuscript. We agree that VOCs play a key role in the activation of plant immune system or basal defense responses against various biotic stress. However, these volatile compounds can also have affect on the pathogen directly in vitro assay (Kong et al. 2020, Front. Microbiol.; He et al. 2020, Molecules; Kaddes et al. 2019, Int. J. Environ. Res. Public Health; Calvo et al. 2019, Postharvest Biology and Technology). We have included this point in discussion section.
Line 215 ff and Figure 7: The authors should mention; which VOCs were used for the application experiment. Based on Material and Method line 421 I assume that also for this experiment the VOCs from, LA1777 were used, but it is not very clear.
Thanks for the comment. We have added the information in MS, so it should be clearer now.
Line 326 and Table 1: The authors describe the use of the commercial strain ‘Super Strain B’ for VOC analysis. However, the efficacy of VOCs was tested on another susceptible cultivar “Castlerock’. The authors should describe why they used two different cultivars for those experiments.
Both cultivars (“Castlerock’ and Super Strain B) are susceptible to late blight disease based on our previous studies (Arafa et al. 2017, PLoS ONE; Arafa et al. 2017, Afr. J. Agric. Res.). We had very limited seeds from Super Strain B in the second experiment.
Line 396: VOC are metabolites which express their physiological reaction after being emitted from the producing plant. Therefore, most studies on VOCs use air sampled from the immediate vicinity of the plants for collecting VOCs. In the presented study, metabolites were isolated directly from leaflets, which could have an impact on the results and should be explained and discussed.
Thanks for the comment. There are different approaches for VOCs extraction with advantages and disadvantages of each one (Materić et al. 2015, Applications in Plant Sciences). For instance, Son et al. (2021) extracted the VOCs from Ambrosia artemisiifolia L. and Artemisia annua L. leaves by homogenized samples and mixed with distilled water. Antonious, (2016, J Environ Anal Toxicol) extracted the contents of trichomes of Lycopersicon hirsutum f. glabratum accession PI 134417 through soaking of leaves in water containing 1% Alkamuls that revealed spider mites and aphids. Based on our microscopic observations, the sporangia germination, appressorium formation and the mesophyll cells penetration were very slow or absent in late blight resistant wild accession LA1777. We expect the leaves of LA1777 might produce some volatiles that prohibit the progression of the pathogen on plant leaf surface. In our present study, we extracted VOCs by methylene chloride, which was used in previous studies ((Tieman et al. 2006, PNAS; Mathieu et al. 2009, J. Exp. Bot.; Goulet et al. 2015, Molecular Plant; Wang and Kays, 2000 from sweet potato J. AMER. SOC. HORT. SCI.,).
Line 422: The method is not described clearly. ‘The prepared solution was amended with rye agar medium 422 and inoculated with 3 mm mycelium disk’. More details needed.
Thanks for the comment. We have added information requested in section 4.5
The authors should double-check the manuscript for minor mistakes with the English language and some typos.
Thanks for the comment. The manuscript has been edited by native English speaker and some typos were modified.
Reviewer 2 Report
I reviewed the manuscript "Leaf Extracts from Resistant Wild Tomato Can Be Used to Control Late Blight (Phytophthora infestans) in Cultivated Tomato" submitted to the journal Plants.
The idea of the manuscript is interesting, but I think the experiment, especially the GC-MS analysis, was not done properly. From the Material and Method section and the figures, it appears that the authors did not perform the analysis in replicates, which could affect the credibility of the results. In metabolomic studies, at least five biological replicates are recommended because the possibility of biological variation between samples is high. If only one analysis is performed, we do not know whether it is the influence of the variable under study or random variation.
My other comments are:
L 38-39 "Worldwide, tomato (Solanum lycopersicum L.) is the second most important vegetable crop after potato (S. tuberosum L.)." Please add the reference or delete the statements. What does the most important mean? The most commonly grown? Used?
Figure 8 is blurry, should be larger.
Figure 2, 3 and - Did you perform your experiments in replicates? If yes, please add the standard deviations. Unfortunately, if you did not perform the analysis in replicates, it cannot be published that way. Proper replications are one of the most important steps in experiments, especially in metabolomic studies where there can be large variations.
Conclusions are some general statements, without information from the manuscript.
Author Response
June 07, 2022
Dear,
We have attached our revised manuscript, " Leaf Extracts from Resistant Wild Tomato Can be Used to Control Late Blight (Phytophthora infestans) in Cultivated Tomato”, which is submitted to Plants Journal (plants-1756720).
After careful consideration of the reviewers’ comments, we made a number of major revisions and improvements. The revised manuscript is submitted with using the Track Changes function in Word but we have also attached the same document with all changes accepted so it is easier to read
Below is a detailed feedback to the comments. Our response is in bold.
Reviewer 2
I reviewed the manuscript "Leaf Extracts from Resistant Wild Tomato Can Be Used to Control Late Blight (Phytophthora infestans) in Cultivated Tomato" submitted to the journal Plants.
The idea of the manuscript is interesting, but I think the experiment, especially the GC-MS analysis, was not done properly. From the Material and Method section and the figures, it appears that the authors did not perform the analysis in replicates, which could affect the credibility of the results. In metabolomic studies, at least five biological replicates are recommended because the possibility of biological variation between samples is high. If only one analysis is performed, we do not know whether it is the influence of the variable under study or random variation.
Thanks for the comment. We used three replicates for the GC-MS analysis. We are sorry for not having mentioned this in the first round but have now made this clear, first time in the abstract where we now say: “The metabolic profiles were obtained in both inoculated and non-inoculated plants by analyzing leaf extracts using high-resolution gas chromatography-mass spectrometry (GC-MS) with three replicate analysis of each genotype.”; and thereafter in the Material and Methods section where we now say: “Volatile organic compounds (VOCs) analysis was done on all five genotypes, and with three replicate analysis per genotype.” See below specific replies to your comments.
My other comments are:
L 38-39 “Worldwide, tomato (Solanum lycopersicum L.) is the second most important vegetable crop after potato (S. tuberosum L.).” Please add the reference or delete the statements. What does the most important mean? The most commonly grown? Used?
Thanks for the comment. We clarified the statement to be “commonly grown”, and a reference has been added. Thereafter we refer to the FAO statistics with concrete production numbers.
Figure 8 is blurry, should be larger.
Thanks for the comment. The quality of the figure has now been improved an should be easier to read.
Figure 2, 3 and 4 - Did you perform your experiments in replicates? If yes, please add the standard deviations. Unfortunately, if you did not perform the analysis in replicates, it cannot be published that way. Proper replications are one of the most important steps in experiments, especially in metabolomic studies where there can be large variations.
Thanks for the comment. We used three replicates for the GC-MS analysis. Figure 2, 3 and 4 have been improved by adding standard deviation on each of the bars.
Conclusions are some general statements, without information from the manuscript.
Thanks for the comment. We have updated the conclusion with key information based on our research. For example we now have included that 31 VOCs were identified in this study and where the wild tomato, accession LA 1777, recorded the highest numbers of VOCs and that application of VOCs in whole plant assay decreased the late blight disease severity compared to control (untreated plants). We add more information on the implications of our research and perspectives for further studies.
Reviewer 3 Report
This article aimed to examine the metabolic profiles in the leaf tissue of late blight resistant wild tomato and to investigate if leaf extracts from such genotypes could be used to control of late blight in tomato production. Before recommending this article for publication, there are some shortcomings for that should be resolve.
Abstract
In abstract no need of literature review such as line 19 and 20.
“compounds sfrom’ check typos.
Introduction
The introduction part is well written but still some details are required.
Economic and medicinal importance of tomato. The following articles may be cited.
https://doi.org/10.1016/j.micpath.2020.103966, https://doi.org/10.1007/s10725-021-00785-7,
The authors must discuss genes and pathways involve in antimicrobial resistance
Results
In figure 6, 7 mention the plants which one is control and VOC.
Methods
In workflow text must be clear.
Microscopic identification section should be cited with the following article.
https://doi.org/10.1016/j.bcab.2020.101729,
Conclusion
Conclusion is well justified.
Author Response
Dear,
We have attached our revised manuscript, " Leaf Extracts from Resistant Wild Tomato Can be Used to Control Late Blight (Phytophthora infestans) in Cultivated Tomato”, which is submitted to Plants Journal (plants-1756720).
After careful consideration of the reviewers’ comments, we made a number of major revisions and improvements. The revised manuscript is submitted with using the Track Changes function in Word but we have also attached the same document with all changes accepted so it is easier to read
Below is a detailed feedback to the comments. Our response is in bold.
Reviewer 3
This article aimed to examine the metabolic profiles in the leaf tissue of late blight resistant wild tomato and to investigate if leaf extracts from such genotypes could be used to control of late blight in tomato production. Before recommending this article for publication, there are some shortcomings for that should be resolve.
Abstract
In abstract no need of literature review such as line 19 and 20.
Thanks for the comment. We agree with you, the sentence was deleted.
“compounds sfrom’ check typos.
Corrected.
Introduction
The introduction part is well written but still some details are required.
Economic and medicinal importance of tomato. The following articles may be cited.
https://doi.org/10.1016/j.micpath.2020.103966, https://doi.org/10.1007/s10725-021-00785-7,
Thanks for the comment. The requested references and information have now been added to introduction section, section 1. The new references are included in the reference list.
The authors must discuss genes and pathways involve in antimicrobial resistance
Thanks for the comment. This point is now included, see section 1.
Results
In figure 6, 7 mention the plants which one is control and VOC.
Thanks for the comment. In Figure 6. A and C present a positive control (sprayed by P. infestans only) three and seven days post inoculation, respectively. B and D present infected plants by P. infestans and treated by VOCs where complete suppression and no symptoms of tomato late blight.
In Figure 7. Left row refers to sprayed plants by P. infestans and VOCs whereas the right row refers to infected plants (positive control).
Methods
In workflow text must be clear.
Thanks for the comment. The workflow text in Fig 8 has been improved with font and size changes.
Microscopic identification section should be cited with the following article.
Thanks for the comment. The provided reference : https://doi.org/10.1016/j.bcab.2020.101729” is now cited in the manuscript, line 449, section 4.3.
Round 2
Reviewer 2 Report
Still aithors say three replicate analysis. What I want to point out that they need at least 3 biological replicates, not techical (three analysis).